# Temporal Trend of Conventional Sperm Parameters in a Sicilian Population in the Decade 2011–2020

**DOI:** 10.3390/jcm10050993

**Published:** 2021-03-02

**Authors:** Rossella Cannarella, Rosita A. Condorelli, Carmelo Gusmano, Nunziata Barone, Nunziatina Burrello, Antonio Aversa, Aldo E. Calogero, Sandro La Vignera

**Affiliations:** 1Department of Clinical and Experimental Medicine, University of Catania, 95123 Catania, Italy; rossella.cannarella@phd.unict.it (R.C.); rosita.condorelli@unict.it (R.A.C.); carmelo.gusmano@yahoo.it (C.G.); baronen@unict.it (N.B.); burrello@policlinico.unict.it (N.B.); acaloger@unict.it (A.E.C.); 2Department of Experimental and Clinical Medicine, Magna Græcia University, 88100 Catanzaro, Italy; aversa@unicz.it

**Keywords:** sperm concentration, total sperm count, sperm progressive motility, decline, infertility

## Abstract

Objective: To evaluate the changes of conventional sperm parameters in men who referred to an andrology reference center in Catania (Eastern Sicily, Italy) in the decade 2011–2020. Methods: For this purpose, we selected–retrospectively and randomly–the reports of 1409 semen analyses performed according to the 2010 WHO criteria. Data on sperm concentration, total sperm count, progressive sperm motility, and percentage of normal forms were analyzed using linear regression of the raw and logarithmic-transformed data. The sperm parameters were subsequently pooled in two five-year periods (2011–2015 and 2016–2020) and compared with each other. Finally, the influence of the city of residence was assessed on five-year pooled data. Main results: A slight but non-significant decline of total sperm count (−2.26 million/year; *p* = 0.065) and the percentage of spermatozoa with normal morphology (−0.08%/year; *p* = 0.057) was observed. In contrast, a significant increase of progressive sperm motility (+0.28%/year; *p* = 0.008) over time was found. The total sperm count of the quinquennium 2016–2020 was significantly lower. and an upward trend of progressive sperm motility was found. compared to the years 2011–2015. No changes in sperm concentration and morphology occurred in the years 2011–2015 vs. 2016–2020. Sperm conventional parameters did not differ when the five-year pooled data were analyzed according to the town of residence. Conclusions: Divergent trends of total sperm count and progressive sperm motility over time were found in patients from Eastern Sicily. This may point out the need of assessing whether a time-dependent change of biofunctional sperm parameters occurs to really understand the trend of sperm quality over time.

## 1. Introduction

Whether a temporal trend toward a decline of the sperm quality really exists is still a matter of debate, since both supporting [1,2,3,4,5,6,7,8,9,10,11,12,13,14,15] and opposing [16,17,18,19,20,21,22,23,24,25] evidence has been published so far. Two meta-regression studies collecting data from hundreds of people from several countries have been carried out in an attempt to resolve conflicting evidence. The first analyzed data from nearly 43,000 men showed a decline of 52.4% of sperm concentration and 59.3% of total sperm count between 1973 and 2011 among unselected men living in Western countries (North America, Europe, Australia, and New Zealand).

Notably, such a decline was not observed in men from South America, Asia, and Africa [26]. However, a second meta-analysis carried out on the African population reported a time-dependent decrease by 72.6% in the past 50 years (1965–2015) [27]. Furthermore, the presence of a sperm-parameter decline throughout the decades has also been described in Asian countries [28,29] (Appendix A), thus contradicting the findings of the first meta-analysis and making arduous the interpretation of the results. 

Numerous hypotheses have been put forward to explain the controversial results obtained so far. Among these, geographical and temporal variability, the presence of possible confounding factors, or the heterogeneity of the enrolled cohorts [30] have been suggested. The importance of solving this disagreement is reinforced by the observation that sperm output can predict mortality and morbidity (in addition to fertility, albeit with some limitations) of a given population [31]. Therefore, understanding if there is really a decline in sperm production could be useful to gain further insights into the general health of a specific cohort of people.

Even the studies that attempted to evaluate the trend of sperm quality in Italy alone have shown mixed results. In particular, one study described a 30.7% decreased mean sperm concentration between 1981 and 1995, as well as for total motility and morphology. Nevertheless, no statistical analysis was performed making these data of little significance [32]. Data from south-eastern Sicily suggest the existence of an over-time decline in sperm concentration and normal morphology from 1982 to 1999 [33]. On the other hand, a comparative study analyzed the sperm parameters of 701 men collected in 1992 with those of 626 men collected in 2010, describing an improvement in sperm concentration, total sperm count, and progressive sperm motility [34]. However, these studies are influenced by the use of the different WHO guidelines published over the years (1980, 1987, 1992, 1999, and 2010). Moreover, some studies have compared data collected from just two different years [32,34], rather than analyzing the existence of a trend over time. 

No recent data are available on the drift of sperm quality in Italy. Therefore, the purpose of this study was to evaluate whether there have been changes in conventional sperm parameters in the decade starting in 2011, the first year after the publication of the last edition of the WHO manual for performing semen analysis [35]. To achieve this goal, randomly selected sperm analyses were retrospectively analyzed in a national reference center for andrology, located in Catania, Eastern Sicily, Italy, where patients refer for andrological control, fertility, varicocele, male accessory gland infection, follow-up of varicocelectomy, cryptorchidism, and other andrological concerns, as well as upon request by other physicians. 

## 2. Materials and Methods

### 2.1. Patient Selection

This is a longitudinal cohort retrospective study conducted on 1409 randomly selected seminal fluid analyses of men who were referred to the Seminology Laboratory of the Division of Endocrinology, Metabolic Diseases and Nutrition, University-Teaching Hospital “G. Rodolico”, University of Catania, from January 2011 to January 2020. These medical reports were extracted using a random criterion from the total number of seminal fluid exams performed by the Seminology Laboratory (~9000 from 2011 to 2020).

In line with previous studies, which have included 10% of the overall sperm analyses [2,22], we decided to include 20% of the sperm analyses performed each year. The specimens were randomly selected. To accomplish this, the medical reports available for each year were sorted alphabetically based on the patient’s surname (from A to Z) and chosen according to the number of their position selected by the randomization software. After inclusion, the medical reports of the same patients were excluded. Thus a total of 1409 sperm analyses were obtained.

No exclusion criterion was used to make the results generalizable to the male population and avoiding the restriction due to specific conditions. In addition to the semen analysis, the clinical history, including age and city of residence, was collected from each man, although the clinical data of patients followed by other physicians were not available.

The study has been conducted according to the principles expressed in the Declaration of Helsinki.

### 2.2. Sperm Analysis

The sperm analyses were performed by the same two operators who received the same training and professional update and used the same methods throughout the 10 years. Both external quality control (EQC) and internal quality control (IQC) were carried out, as indicated in the 2010 WHO manual [35], and the absence of significant difference in the external and internal analysis was ascertained. EQC was performed every 6 months by comparing the sample analysis made at the Seminology Laboratory of the Division of Endocrinology, Metabolic Diseases, and Nutrition, with that performed by the Quality Control Team of the Central Laboratory of the University Teaching-Hospital Policlinico “G. Rodolico-San Marco”. The IQC was carried out every 6 months by making replicate measurements on separate aliquots of a semen sample by the two technicians of our Seminology Laboratory. The warning and action control limits were estimated using an S-chart plot 30. Calibration of pipettes, counting chambers, and the assessment of the other equipment was performed yearly [35]. 

Semen samples were collected by masturbation into a sterile container after 2–7 days of sexual abstinence and were analyzed immediately after liquefaction. Each sample was evaluated for seminal volume, pH, total sperm count, progressive motility, morphology, and leukocyte concentration [35]. Semen volume was measured by graduated pipettes. Calibration strips were used to measure the seminal fluid pH. 

For the evaluation of the sperm concentration, following semen liquefaction, 10 µL of non-diluted, well-mixed semen sample was at first loaded in the middle of a clean Neubauer counting chamber, maintained at the temperature of 37 °C, gently covered with a cover glass, and examined using 200× or 400× magnification. The sample was diluted before proceeding with the sperm count. The amount of dilution was decided after the evaluation of the undiluted sample. In particular, when the seminologist found >101 (400×) or >404 (200×) spermatozoa per microscopic field, a 1:20 dilution was used. When the number of spermatozoa was 16–100 (400×) or 64–400 (200×), a 1:5 dilution was used. When the number was 2–15 (400×) or 8–60 (200×), a 1:2 dilution was used. Finally, when the number was <2 (400×) or <8 (200×), a 1:2 dilution was used, strictly following the WHO 2010 manual recommendations [35]. The final concentration was calculated as: [(number of spermatozoa counted/the number of lines) x dilution factor] and expressed as 10^6^ spermatozoa/mL. 

To evaluate the sperm motility, immediately after semen liquefaction, 10 µL of undiluted, well-mixed semen sample was loaded in the middle of a clean Neubauer counting chamber, maintained at the temperature of 37 °C, gently covered with a cover glass, and examined using 200× magnification. Sperm motility was assessed in 200 random spermatozoa and characterized as progressive and non-progressive motility. The total motility was calculated as the sum of progressive and non-progressive motility. Both progressive and total motility were expressed as percentages.

Sperm morphology was evaluated using the Papanicolaou staining procedure. To this end, morphology was evaluated in 200 spermatozoa and the value was expressed as percentages. Finally, the vitality was assessed using eosin staining according to the WHO 2010 manual [35] recommendations. Semen leukocytes were evaluated by the peroxidase test to distinguish them from the other round cells (e.g., immature germ cells) [35]. 

### 2.3. Statistical Analysis

The normality of the variables was evaluated with the Shapiro–Wilks test. Following the analysis of data distribution, the descriptive statistic was presented as the median. The 25th–75th interquartile range for skewed variables, and the mean ± standard deviation (SD) for normally distributed variables. Correlation analysis was evaluated using the Spearman rank correlation and the results have been reported as correlation coefficient (r) and *p*-value. Linear regression analysis was used to examine trends of sperm conventional parameters over time. Due to the skewed distribution, data were log-transformed before being analyzed by linear regression. The univariate model was adopted. Semen parameters were considered as dependent variables and the year as the independent variable. The median values of the conventional sperm parameters of the years 2011–2015 were pooled together and compared with those of the years 2016–2020 using the Wilcoxon signed-rank test. The data were also compared according to the town of residence (Catania vs. other areas) using the Wilcoxon signed-rank test. The statistical analysis was performed using SPSS 22.0 for Windows (SPSS Inc., Chicago, IL, USA). Statistical significance was accepted when the *p*-value was lower than 0.05.

## 3. Results

The prevalence of patients with varicocele, cryptorchidism, and urogenital infection did not differ significantly over the years, although clinical information was not available for all of the men included in this study. Age and conventional sperm parameters for each year (2011–2020) are shown in Table 1. The age of the men enrolled in this study did not show any significant difference over time. Linear regression analysis of the raw data revealed that sperm concentration did not change over time, but total sperm count showed a downward trend (−2.26 million/year; *p* = 0.065). Semen volume showed a significant decline through decades, whereas seminal fluid pH significantly increased, although it remained in the normal range (Table 1). Similarly to the total sperm count, a downward trend was observed for the percentage of spermatozoa with normal morphology over time (−0.08%/year; *p* = 0.057). In contrast, progressive sperm motility increased significantly over time (+0.28%/year; *p* = 0.008) (Figure 1). Linear regression analysis of the log-transformed data confirmed no changes in sperm concentration, total count, and morphology over time. The significant amelioration of progressive sperm motility over the years was also confirmed by this analysis (Table 2). 

The total sperm count of the years 2016–2020 was significantly lower compared to those of the years 2011–2015 (*p* = 0.03) and a trend toward a higher sperm progressive motility in the years 2016–2020 was found compared to 2011–2015 (*p* = 0.058). Sperm concentration and the percentage of spermatozoa with normal morphology were similar (Figure 2). Finally, no difference was found when data were grouped according to the town of residence (Table 3).

## 4. Discussion

This study, carried out in a cohort of randomly selected patients who requested a sperm analysis in an andrological reference center in Catania (Eastern Sicily, Italy), reports a downward trend of total sperm count and the percentage of spermatozoa with normal forms over time. Interestingly, we also found that progressive sperm motility increased significantly in the same decade of observation (2011–2020). Furthermore, the total sperm count pooling together the tests done in the 2016–2020 quinquennium was significantly lower compared to that of the five-year period 2011–2015, while an upward trend for progressive sperm motility and no change for sperm concentration and morphology were found.

These results lead to not-univocal interpretations. First, the significantly lower total sperm count found in the quinquennium 2016–2020, compared to that of the years 2011–2015, is in line with a meta-regression study of the European population [26]. This may negatively impact fertility. Indeed, the fecundity rate in Italy has passed from 2.3 in 1952 to 1.3 in 2019 (http://demo.istat.it/fecondita/index.html (accessed on 23 February 2021)), although this data might be due also to the change of social customs and, in particular, to the delayed time of conception. We did not find a significant decline, only a declining trend, of the total sperm count through years, probably due to the relatively low sample size. It is noteworthy that the semen volume also decreased significantly over time, and this outcome could explain the lack of difference in the total sperm count. Moreover, a significant improvement of sperm progressive motility was found, which appears as contradictory data. On this account, it must be considered that motility is an operator-dependent parameter. However, in the attempt of limiting this confounding factor, the same two operators who received the same training and professional updates performed the sperm analysis throughout the 10 years taken into consideration.

Previous studies have linked the decline in sperm parameters to environmental changes that have occurred through decades. Particularly, endocrine disruptors, heavy metals, chemicals, and lifestyle factors can impact testicular development and function starting from a prenatal age. As an example, fetal exposure to bisphenol-A, a chemical with estrogenic activity, disrupted spermatogenesis damaging viability, motility, and sperm chromatin condensation of adult mice [36]. The negative impact of this chemical on human fertility has been already ascertained [37], as well as that of phthalates [38]. Similarly, fetal exposure to cigarette smoke in mice led to a dose-dependent decrease of epididymal sperm counts both in F1 and F2 male offspring, indicating a transgenerational deleterious effect [39]. Additionally, several lines of evidence address to cigarette smoke a damaging effect on human sperm quality by interacting with the sperm nicotine receptor [40,41]. These data could also explain the worryingly high prevalence of low testicular volume (<12 mL), which is directly related to sperm output [42], registered in high-school Italian students. It has been estimated to be as high as 14% and relates to health-risk behaviors. These include smoking and the use of drugs and alcohol [43]. The decrease in sperm count through the years and the high prevalence of testicular hypotrophy recently registered may therefore represent two faces of the same coin, likely reflecting increasing environmental contamination. However, this evidence is in stark contrast to the improvement of sperm progressive motility, which was analyzed for all the years of the decade using the same WHO criteria [35]. This is a surprising finding that has rarely been reported. Indeed, the main studies show that motility also declines over time [8,28,42,44].

No immediate hypothesis can be proposed to explain the significantly lower total sperm count between the quinquennium 2016–2020 compared to that 2011–2015 five-year periods and the concomitant improvement of progressive sperm motility which appears as contradictory data. Sperm motility is influenced by seminal reactive oxygen species (ROS), which derive from several conditions, including urogenital infection. However, urogenital infections were reported with an extremely low frequency in the cohort of men analyzed, and no significant difference in their prevalence was found among years (Table 1). This makes the role of seminal infections in the amelioration of progressive sperm motility unlikely. A significant increase in seminal pH was observed (although it remained within the normal range). The reasons for this finding are not immediately explainable and its role in the improvement of sperm progressive motility is not known. This highlights the importance of evaluating bio-functional sperm parameters (e.g., sperm chromatin compactness, mitochondrial function, DNA fragmentation) which, although not completely standardized [45], are useful for studying male fertility more closely. In this regard, the guidelines for recurrent pregnancy loss suggest evaluating sperm DNA fragmentation [46]. As a result, conventional sperm parameters, although capable of providing information on fertility potential, may be normal in patients unable to impregnate their partners, leading to the observation that other molecular factors must be taken into consideration to better evaluate the sperm quality [47,48]. 

Therefore, the evaluation of time-dependent changes in bio-functional sperm parameters and their possible association with morbidity and mortality would be very promising. Furthermore, progressive sperm motility plays a key role in human fertilization and directly correlates with sperm mitochondrial function. Mitochondria represent the energy source of spermatozoa and their function is important for sperm hyperactivation, capacitation, acrosomal reaction, and fertilization [49]. Given the significant increase in progressive sperm motility of spermatozoa over the years, it would be interesting to evaluate the time-dependent change of sperm mitochondrial function.

Our analysis also took into consideration the residence of the cohort of men studied. Interestingly, the conventional sperm parameters of men living in the city of Catania showed no significant difference compared to men living in cities smaller than Catania and rural areas. This lack of difference may be due to the fact that the inhabitants of both industrial and agricultural areas are in any case exposed to spermotoxic substances, such as pollutants for industrial areas [50] or pesticides [51]. Furthermore, it could be assumed that at least a part of the men who live outside Catania were commuters; therefore, they too were exposed to a greater degree of urban pollution [52].

The results of the present study must be interpreted with caution, due to the study’s limitations. Compared to some previous studies on this topic (Appendix A), the cohort of men is limited. However, the sperm analyses included were randomly selected from all those performed in the decade 2011–2020, as reported in previous studies [2,18]. Furthermore, no exclusion criterion was used. Although this choice was done to make the results generalizable to the male population, so that restriction to a specific clinical condition could be avoided, we recognize that this can limit the interpretation of the results. Furthermore, the absence of confounding factor analysis, the lack of data on alcohol, smoking, or drug use, and the presence of obesity or other comorbidities (other than varicocele, history of cryptorchidism, and urogenital infections) are potential weaknesses in this study. On the other hand, the longitudinal analysis, no changes in the laboratory staff, the use of the WHO 2010 guidelines for all the semen analysis performed, and the random selection of patients are the strengths of the present study, which was assessed on a representative sample of the Sicilian population. Moreover, this is the first study investigating the temporal trends of sperm conventional parameters using exclusively the criteria reported by the WHO 2010 manual. Recently, the importance of assessing the real applicability of these criteria in 2021 has been pointed out [53]. Therefore, the results of this study may be useful also for this aspect.

In summary, these data indicate the presence of a not statistically significant declining trend in the total sperm count, in the percentage of sperm with normal morphology, and a significant increase of progressive sperm motility in a cohort of 1409 semen tests randomly selected during the last decade (2011–2020) in a highly specialized andrology center in Catania (Eastern Sicily, Italy). Furthermore, in the 2016–2020 five-year period, the total sperm count was significantly lower than in the 2011–2015 five-year period. An upward trend in progressive sperm motility and no change in sperm concentration and morphology were also observed. Therefore, these results seem to be somewhat conflicting due to the opposite tendency of total sperm count (significant lower total sperm count between the quinquennium 2016–2020 compared with that 2011–2015) and progressive sperm motility, and thus show that the evaluation of conventional sperm parameters alone is not able to fully evaluate the quality of the sperm. This suggests the need to also evaluate the changes over time of the biofunctional sperm parameters to provide a more complete understanding of the sperm quality trend over decades.

## Figures and Tables

**Figure 1 jcm-10-00993-f001:**
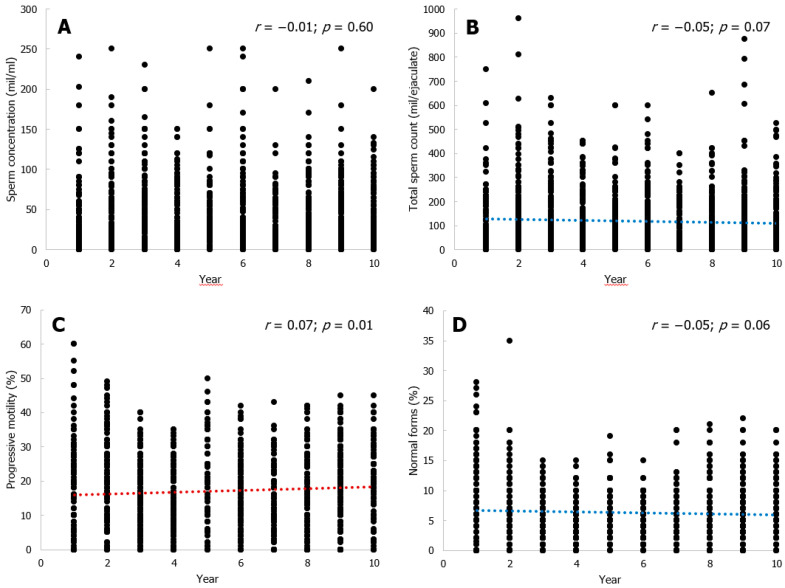
Conventional sperm parameters in a Sicilian cohort of men in the decade 2011–2020. No trend for sperm concentration was observed (panel (**A**)). Total sperm count (panel (**B**)) and the percentage of spermatozoa with normal morphology (panel (**D**)) showed a downward trend over time. Sperm progressive motility increased significantly over time (panel (**C**)). The dotted lines represent the regression lines.

**Figure 2 jcm-10-00993-f002:**
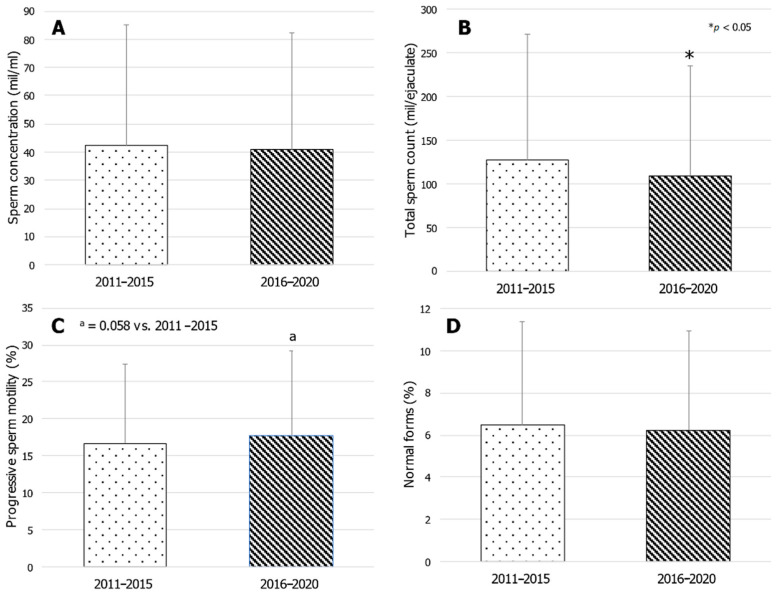
Analysis of conventional sperm parameters after pooling the data into five-year periods (2011–2015 vs. 2016–2020). Sperm concentration (panel (**A**)) and morphology (panel (**D**)) showed no difference. The total sperm count of the years 2016–2020 was significantly lower than that of 2011–2016 (panel (**B**)). Progressive sperm motility showed an upward trend in the years 2016–2020 compared to 2011–2015 (panel (**C**)).

**Table 1 jcm-10-00993-t001:** Age (mean ± DS) and conventional sperm parameters (median and 25th–75th interquartile range) from 2011 and 2020.

Year (*n*)	Age (Years)	Volume (mL)	pH	Sperm Concentration (×10^6^/mL)	Total Sperm Count (×10^6^/ejaculate)	Progressive Sperm Motility (%)	Sperm Normal Forms (%)	Semen Leukocytes (×10^6^/mL)
2011 (135)	31.81 ± 8.05	3.0 (2.0–4.0)	7.9 (7.6–8.1)	14.0 (3.0–45.0)	40.0 (7.0–153.0)	16.0 (6.0–25.0)	10.0 (5.0–14.0)	0.45 (0.07–1.0)
2012 (141)	34.06 ± 7.15	3.2 (2.5–4.5)	8.0 (7.8–8.1)	30.0 (10.0–62.0)	95.0 (27.0–210.0)	17.0 (10.0–27.0)	6.0 (3.0–10.0)	0.5 (0.05–1.0)
2013 (170)	32.24 ± 9.30	3.0 (2.0–4.0)	8.0 (7.8–8.1)	35.0 (8.0–69.0)	75.0 (21.6–220.0)	15.0 (8.0–21.0)	5.0 (2.0–8.0)	0.54 (0.08–1.1)
2014 (146)	32.24 ± 9.34	3.0 (2.0–4.0)	8.0 (7.9–8.2)	27.5 (7.0–70.0)	74.3 (25.5–195.0)	12.0 (7.0–20.0)	4.0 (2.0–6.0)	0.47 (0.06–1.0)
2015 (103)	32.80 ± 9.34	2.5 (2.0–3.8)	8.0 (7.9–8.2)	40.0 (14.0–70.0)	105.6 (39.8–175.0)	19.0 (10.0–28.0)	5.0 (3.0–8.0)	0.72 (0.35–1.4)
2016 (136)	34.13 ± 8.77	2.5 (2.0–3.5)	8.1 (7.9–8.2)	40.0 (16.5–80.0)	103.0 (47.0–200.0)	15.5 (8.5–24.0)	5.0 (2.5–8.0)	0.8 (0.31–1.5)
2017 (127)	33.89 ± 9.63	2.5 (1.7–3.5)	8.1 (7.9–8.2)	20.0 (5.0–57.0)	50.0 (14.0–130.0)	12.0 (6.0–20.0)	5.0 (2.0–8.0)	0.46 (0.18–1.2)
2018 (154)	32.75 ± 8.59	2.5 (1.5–3.5)	8.1 (8.0–8.3)	30.0 (6.0–60.0)	72.5 (15.0–135.0)	18.0 (10.0–26.0)	5.0 (3.0–8.0)	0.53 (0.2–1.2)
2019 (182)	32 (26–39)	2.6 (1.5–4.0)	8.1 (7.9–8.2)	25.0 (6.0–60.0)	66.0 (13.5–156.0)	20.0 (10.0–30.0)	6.0 (4.0–9.0)	0.52 (0.13–1.2)
2020 (115)	31.53 ± 8.64	3.0 (2.0–4.0)	8.1 (7.9–8.1)	32.0 (11.0–70.0)	81.0 (27.0–196.0)	20.0 (14.0–28.0)	7.0 (4.0–11.0)	0.6 (0.18–1.1)

**Table 2 jcm-10-00993-t002:** Linear regression analysis of log-transformed data.

Parameter	R^2^	*β*	*P*
Sperm concentration	0.001	0.026	0.34
Total sperm count	0.000	−0.007	0.80
Progressive sperm motility	0.120	+0.108	<0.00
Spermatozoa with normal morphology	0.001	−0.240	0.39
Semen volume	0.009	−0.09	<0.00
pH	0.044	0.21	<0.00
Leukocytes	0.002	0.53	0.05

**Table 3 jcm-10-00993-t003:** Conventional sperm parameters after pooling into two five-year periods (2011–2015 and 2016–2020) of men living in Catania vs. those who live in other towns of Eastern Sicily (Italy).

Parameter	2011–2015		2016–2020	
	Catania (*n* = 227)	Other Towns (*n* = 217)	Catania (*n* = 259)	Other Towns (*n* = 249)
**Age**	31.0 (25.0–38.0)	33.0 (27.3–39.0)	31.0 (26.0–38.0)	35.0 (28.0–40.0)
**Sperm concentration (×10^6^/mL)**	42.0 (9.0–75.0)	30.0 (10.0–65.0)	35.0 (12.0–70.0)	30.0 (6.0–62.5)
**Total sperm count (×10^6^/ejaculate)**	115.0 (24.0–220.0)	75.0 (27.5.0–210.0)	81.0 (25.2–175.0)	81.0 (17.5–168.0)
**Progressive sperm motility (%)**	15.0 (8.0–24.0)	15.0 (8.0–25.0)	18.0 (10.0–26.0)	18.0 (10.0–25.0)
**Sperm normal forms** **(%)**	6.0 (3.0–9.0)	5.0 (3.0–8.0)	6.0 (3.0–9.0)	5.0 (3.0–9.0)

## Data Availability

Data are available upon request to the corresponding author.

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
