# Peer review of "Temporal Trend of Conventional Sperm Parameters in a Sicilian Population in the Decade 2011–2020"

_jcm, 2021, doi:10.3390/jcm10050993_

Round 1
Reviewer 1 Report
In the manuscript titled “Temporal trend of conventional sperm parameters in a Sicilian population in the decade 2011-2020" the authors evaluated the changes in conventional sperm parameters in men visiting an Andrology reference center in Eastern Sicily, Italy within a period of ten years from 2011 to 2020. This is an interesting longitudinal cohort retrospective study and specifically in the epidemiologic field. However, the manuscript needs several significant improvements. Detailed comments are as follows:
- Abstract, conclusion, lines 24-26, the term "opposite trends" is very confusing as it is comparing more than 2 parameters. Please reword and clarify.
- No power analysis was included in the study regarding calculating the sample size. The authors need to explain why only 20% of semen samples were considered to be sufficient for the stated statistical analysis.
- Please explain in detail if the semen samples were diluted when sperm concentration and motility were evaluated.
- Please briefly write the protocols which were used for evaluation of sperm morphology and motility (in sperm analysis section).
- Page, line 142, and table 2, correct “Sperm volume” as semen volume, please review the entire manuscript as semen and/or sperm were used interchangeably with no consistency.
- Semen volume can be affected by multiple factors especially the abstinence day, which was not the same in all samples, it could not be concluded that semen volume has decreased over the years. The authors reported no change in sperm concentration which does not rely on volume. Please emphasize in the manuscript.
- Motility assessment using manual microscopy method is solely operator-dependent and relies on his/her expertise and interpretation and since the authors evaluated a single semen sample from each patient, a clarification is required in the discussion.
- The authors reported leukocytes values without explaining the method used or was it only round cells? Please clarify and change to either "Leukocytes" or "semen leukocytes" as they are not in the sperm cells.
- Calculating and reporting "Total Motile Sperm Count" is important for the clinical application and will be of value if included in the revised manuscript.
Author Response
Answers to Reviewer #1 comments
Manuscript ID jcm-1105383
Comment 1: Abstract, conclusion, lines 24-26, the term "opposite trends" is very confusing as it is comparing more than 2 parameters. Please reword and clarify.
Answer to comment 1: We referred to the decreased total sperm count and the increased sperm progressive motility. We, therefore, rephrased the sentence as follows: “Divergent trends of total sperm count and progressive sperm motility over time was found in patients from Eastern Sicily”
Comment 2: No power analysis was included in the study regarding calculating the sample size. The authors need to explain why only 20% of semen samples were considered to be sufficient for the stated statistical analysis.
Answer to comment 2: We thank the reviewer for this comment. We acknowledge that we have not describe this point clearly enough in the manuscript. Following the methodology of previous studies, we included 20% of the sperm analyses done each year. Previous studies exploring the temporal trend of sperm conventional parameters have included 10% of the sperm analyses randomly selected among all those made. For example, Adamopoulos and colleagues included 2385 samples on a total of 23850 performed in 17 years (Hum Reprod 1996, 11, 1936–1941). Similarly, Marimuthu and colleagues enrolled 1176 specimens that were 10% of the records analyzed in an 11-year long period (Asian J Androl 2003, 5, 221–225). We apologize for not having explained this clearly in Methods. An explanatory sentence has been now included (please see the section on “Patient selection”).
Comment 3: Please explain in detail if the semen samples were diluted when sperm concentration and motility were evaluated.
Answer to comment 3: This has now been clarified (please see the section “Sperm analysis”).
Comment 4: Please briefly write the protocols which were used for evaluation of sperm morphology and motility (in sperm analysis section).
Answer to comment 4: Done as requested (please see the section “Sperm analysis”).
Comment 5: Page, line 142, and table 2, correct “Sperm volume” as semen volume, please review the entire manuscript as semen and/or sperm were used interchangeably with no consistency.
Answer to comment 5: Done as requested. Thank you.
Comment 6: Semen volume can be affected by multiple factors especially the abstinence day, which was not the same in all samples, it could not be concluded that semen volume has decreased over the years. The authors reported no change in sperm concentration which does not rely on volume. Please emphasize in the manuscript.
Answer to comment 6: Thank you for this observation, which was added in the Discussion (please see page 7).
Comment 7: Motility assessment using manual microscopy method is solely operator-dependent and relies on his/her expertise and interpretation and since the authors evaluated a single semen sample from each patient, a clarification is required in the discussion.
Answer to comment 7: We agree with the reviewer on this point and we added a comment for this aspect (please see page 7). One of the strengths of this study is that the analysis was carried out by the same two seminologists throughout the 10 years. They both received the same training and professional update and used the same methods.
Comment 8: The authors reported leukocytes values without explaining the method used or was it only round cells? Please clarify and change to either "Leukocytes" or "semen leukocytes" as they are not in the sperm cells.
Answer to comment 8: The peroxidase test, as suggested by the WHO 2010 manual, was used to differentiate leukocytes from immature germ cells (please see page 4). “Sperm leukocytes” was corrected. Thank you.
Comment 9: Calculating and reporting "Total Motile Sperm Count" is important for the clinical application and will be of value if included in the revised manuscript.
Answer to comment 9: We thought not to calculate the total motile sperm count as this parameter is not listed among those recommended by the WHO manual for semen analysis whose criteria were strictly adhered to. Furthermore, most studies on this topic do not include this parameter (Supplementary Table 1). Therefore, we would prefer not to include it.
Reviewer 2 Report
Dear Authors
Journal of Clinical Medicine
Manuscript ID: jcm-1105383
Type of manuscript: Article
Title: Temporal trend of conventional sperm parameters in a Sicilian population in the decade 2011-2020
Reviewer comments to Authors:
This manuscript is well written, presents excellent methodology, and covers an important issue in Reproductive Medicine.
However, Authors did not include all articles on the subject, and for that AA should improve their search. It would be interesting to see a Table with all published articles, giving the number of patients studied and the results (N, decreased, increased) for the maim semen parameters (concentration, TSC, TSP, PM).
Bellow I include some of the articles missing:
-Alessa Sugihara, Diane De Neubourg, Usha Punjabi. Is there a temporal trend in semen quality in Belgian candidate sperm donors and in sperm donors’ fertility potential from 1995 onwards? Andrology. 2020. https://doi.org/10.1111/andr.12963
-Serajeddin Vahidi, Mohammad Reza Moein, Fatemeh Yazdinejad, Saeed Ghasemi-Esmailabad, Nima Narimani. Iranian temporal changes in semen quality during the past 22 years: A report from an infertility center. International Journal of Reproductive BioMedicine. 2020; 18 (12): 1059-1064. https://doi.org/10.18502/ijrm.v18i12.8027
-Marcello Cocuzza, Sandro C. Esteves. Shedding Light on the Controversy Surrounding the Temporal Decline in Human Sperm Counts: A Systematic Review. Scientific World Journal. 2014; 365691: 9 pages. http://dx.doi.org/10.1155/2014/365691
-Kasman A, Del Giudice F, Shkolyar E, Porreca A, Busetto GM, Lu Y, Eisenberg M L. Modeling the contribution of the obesity epidemic to the temporal decline in sperm counts. Archivio Italiano Di Urologia E Andrologia. 2020; 92 (4): 357-361. https://doi.org/10.4081/aiua.2020.4.357
-Purusotam Basnet, Sissel A Hansen, Inger K Olaussen, Martha A Hentemann, Ganesh Acharya. Changes in the semen quality among 5739 men seeking infertility treatment in Northern Norway over past 20 years (1993–2012). Journal of Reproductive Biotechnology and Fertility. 2016; 5: 1-7. https://doi.org/10.1177/2058915816633539
-Priyanka Mishra, Mahendra Pal Singh Negi, Mukesh Srivastava, Kiran Singh, Singh Rajender. Decline in seminal quality in Indian men over the last 37 years. Reproductive Biology and Endocrinology. 2018; 16:103. https://doi.org/10.1186/s12958-018-0425-z
-Edson Borges Jr., Amanda Souza Setti, Daniela Paes de Almeida Ferreira Braga, Rita de Cassia Savio Figueira, Assumpto Iaconelli Jr. Decline in semen quality among infertile men in Brazil during the past 10 years. International Brazilian Journal of Urology. 2015; 41 (4): 757-63. https://doi.org/10.1590/S1677-5538.IBJU.2014.0186
Author Response
Answers to Reviewer #2 comments
Manuscript ID jcm-1105383
Comment 1: Authors did not include all articles on the subject, and for that AA should improve their search. It would be interesting to see a Table with all published articles, giving the number of patients studied and the results (N, decreased, increased) for the main semen parameters (concentration, TSC, TSP, PM).
Below I include some of the articles missing:
- Alessa Sugihara, Diane De Neubourg, Usha Punjabi. Is there a temporal trend in semen quality in Belgian candidate sperm donors and in sperm donors’ fertility potential from 1995 onwards? Andrology. 2020. https://doi.org/10.1111/andr.12963
- Serajeddin Vahidi, Mohammad Reza Moein, Fatemeh Yazdinejad, Saeed Ghasemi-Esmailabad, Nima Narimani. Iranian temporal changes in semen quality during the past 22 years: A report from an infertility center. International Journal of Reproductive BioMedicine. 2020; 18 (12): 1059-1064. https://doi.org/10.18502/ijrm.v18i12.8027
- Marcello Cocuzza, Sandro C. Esteves. Shedding Light on the Controversy Surrounding the Temporal Decline in Human Sperm Counts: A Systematic Review. Scientific World Journal. 2014; 365691: 9 pages. http://dx.doi.org/10.1155/2014/365691
- Kasman A, Del Giudice F, Shkolyar E, Porreca A, Busetto GM, Lu Y, Eisenberg M L. Modeling the contribution of the obesity epidemic to the temporal decline in sperm counts. Archivio Italiano Di Urologia E Andrologia. 2020; 92 (4): 357-361. https://doi.org/10.4081/aiua.2020.4.357
- Purusotam Basnet, Sissel A Hansen, Inger K Olaussen, Martha A Hentemann, Ganesh Acharya. Changes in the semen quality among 5739 men seeking infertility treatment in Northern Norway over past 20 years (1993–2012). Journal of Reproductive Biotechnology and Fertility. 2016; 5: 1-7. https://doi.org/10.1177/2058915816633539
- Priyanka Mishra, Mahendra Pal Singh Negi, Mukesh Srivastava, Kiran Singh, Singh Rajender. Decline in seminal quality in Indian men over the last 37 years. Reproductive Biology and Endocrinology. 2018; 16:103. https://doi.org/10.1186/s12958-018-0425-z
- Edson Borges Jr., Amanda Souza Setti, Daniela Paes de Almeida Ferreira Braga, Rita de Cassia Savio Figueira, Assumpto Iaconelli Jr. Decline in semen quality among infertile men in Brazil during the past 10 years. International Brazilian Journal of Urology. 2015; 41 (4): 757-63. https://doi.org/10.1590/S1677-5538.IBJU.2014.0186
Answer to comment 1: We appreciated this comment. All the articles suggested were of great interest for this study and were included except for Cocuzza et al., 2014 and Kasma et al., 2020, which are review articles with no quantitative analysis. The results of articles reporting data on conventional sperm decline over time are now summarized in Supplementary Table 1.
Reviewer 3 Report
I this retrospective study authors evaluate the spermograms in the Region of Catania showing the opposite trends over the time. The study is interesting however the group of patients is not characterized. Probably men were form infertility couples (as I suppose) and there was no control. - that's the limitation of this study - should be mentioned in discussion.
194. Fecundity rate decline in EU because of social issues like delaying conception rather than quality of sperm - look at growing population worldwide.
Author Response
Answers to Reviewer #3 comments
Manuscript ID jcm-1105383
Comment 1: In this retrospective study authors evaluate the spermograms in the Region of Catania showing the opposite trends over the time. The study is interesting however the group of patients is not characterized. Probably men were form infertility couples (as I suppose) and there was no control. - that's the limitation of this study - should be mentioned in discussion.
Answer to comment 1: We thank the reviewer for this comment. We have analyzed randomly selected semen samples throughout a time-span of 10 years and the clinical data of these patients are not available. However, the samples do not come from a cohort made exclusively of infertile patients. It included also patients with varicocele, previous cryptorchidism, urogenital infection, or fertile men wishing to undergo an andrology check-up. This was already recognized in the study limitations. Please see page 9: “the absence of confounding factor analysis, the lack of data on alcohol, smoking or drug use and the presence of obesity or other comorbidities (other than varicocele, history of cryptorchidism, and urogenital infections) are potential weaknesses in this study”. However, the poor clinical characterization of patients is a common aspect of the studies aimed at analyzing the temporal trends of sperm conventional parameters (see Supplementary Table 1). Indeed, the retrospective study design of these analyses makes it unlikely to collect the clinical information of the patients included.
Comment 2: 194. Fecundity rate decline in EU because of social issues like delaying conception rather than quality of sperm - look at growing population worldwide.
Answer to comment 2: This consideration has been added in the Discussion (please see page 7).
Reviewer 4 Report
This paper describes the findings of a retrospective review performed between 2011 and 2020 looking at temporal changes in the WHO criteria for semen analysis. The authors find small declines in total sperm count and normal morphology over time, along with a statistically significant increase in sperm motility. By dividing the ten-year period of analysis into two quintiles, they are able to show significant changes in total sperm count along with the significant increase in sperm motility. The authors included over 1400 data points in the analysis but this represented a randomly selected proportion of 20 percent of the total number of analyses performed in the laboratory in question. They do not explain why they chose to analyse only 20 percent rather than the entire data set available to them. in a Discussion, they include a relatively small size of the cohort as one of the limitations of the study – one therefore has to ask why they did not include more of their available data.
Whilst very relevant to infertility practice in Southern Italy, I question whether this paper contributes significantly to global literature on possible effects of environmental influences and other factors on sperm quality over time. The observed findings are of little clinical significance, although show statistical significance, derived from a fairly small data set. I would encourage the authors to consider analysing a larger proportion of their patients to see whether more significant trends could be observed.
Author Response
Answers to Reviewer #4 comments
Manuscript ID jcm-1105383
Comment 1.
1) The authors included over 1400 data points in the analysis but this represented a randomly selected proportion of 20 percent of the total number of analyses performed in the laboratory in question. They do not explain why they chose to analyze only 20 percent rather than the entire data set available to them.
2) In a Discussion, they include a relatively small size of the cohort as one of the limitations of the study – one Therefore has to ask why they did not include more of their available data.
Answer to comment 1.
1) We have included the 20% of the specimens following an already reported methodology. Indeed, previous studies aimed at assessing the decline of sperm conventional parameters have included 10% of the sperm analyses randomly selected among those of their databases. For example, Adamopoulos and colleagues included 2385 out of 23850 samples evaluated in 17 years (Hum Reprod 1996, 11, 1936–1941). Similarly, Marimuthu and colleagues selected 1176 specimens corresponding to 10% of the records analyzed in an 11-year long period (Asian J Androl 2003, 5, 221–225). This was better written in the revised version of the manuscript (please see page 2).
2) Our sample size is comparable to that of many but not all other studies that have investigated the same topic (please see Supplementary Table 1). To make this concept clear, the sentence has been rewritten as follows: “Compared to some previous studies on this topic (Supplementary Table 1), the cohort of men is limited. However, the sperm analyses included were randomly selected from all those performed in the decade 2011-2020 as reported in previous studies [2,18]”.
Comment 2. Whilst very relevant to infertility practice in Southern Italy, I question whether this paper contributes significantly to global literature on possible effects of environmental influences and other factors on sperm quality over time. The observed findings are of little clinical significance, although show statistical significance, derived from a fairly small data set. I would encourage the authors to consider analysing a larger proportion of their patients to see whether more significant trends could be observed.
Answer to comment 2. As clarified in the answer to the previous comment, we used the same methodology as previous studies (ref. 2, 8). Furthermore, the sample size of the present analysis is comparable to that of a great number of studies on the same topic as detailed in Supplementary Table 1. Moreover, the results of this analysis are in line with previous literature, thus suggesting their reliability. Therefore, we would prefer not to include the data of the overall sample.
Finally, we would like to point out that this is the first study investigating the decline of sperm conventional parameters using exclusively the WHO 2010 manual. The importance of assessing the real applicability of these criteria in 2021 has recently been pointed out [please see ref. 53]. This was included in the revised version of the manuscript. Hence, the present analysis may be useful to this end (this has been added in lines 297-301).
Round 2
Reviewer 1 Report
The authors answered all my questions and edited the manuscript accordingly.
Thank you
Reviewer 4 Report
Unfortunately, the problem of small sample size and non-generalisability remains.